# Nano-imaging photoresponse in a moiré unit cell of minimally twisted bilayer graphene

Niels C. H. Hesp [1], Iacopo Torre [1], David Barcons-Ruiz[1], Hanan Herzig Sheinfux[1], Kenji Watanabe [2], Takashi Taniguchi [3], Roshan Krishna Kumar [1✉] & Frank H. L. Koppens [1,4✉]

Graphene-based moiré superlattices have recently emerged as a unique class of tuneable solid-state systems that exhibit significant optoelectronic activity. Local probing at length scales of the superlattice should provide deeper insight into the microscopic mechanisms of photoresponse and the exact role of the moiré lattice. Here, we employ a nanoscale probe to study photoresponse within a single moiré unit cell of minimally twisted bilayer graphene. Our measurements reveal a spatially rich photoresponse, whose sign and magnitude are governed by the fine structure of the moiré lattice and its orientation with respect to measurement contacts. This results in a strong directional effect and a striking spatial dependence of the gate-voltage response within the moiré domains. The spatial profile and carrier-density dependence of the measured photocurrent point towards a photo-thermoelectric induced response that is further corroborated by good agreement with numerical simulations. Our work shows sub-diffraction photocurrent spectroscopy is an exceptional tool for uncovering the optoelectronic properties of moiré superlattices.

[1] ICFO-Institut de Ciencies Fotoniques, The Barcelona Institute of Science and Technology, Barcelona, Spain. [2] Research Center for Functional Materials, National Institute for Materials Science, Tsukuba, Ibaraki, Japan. [3] International Center for Materials Nanoarchitectonics, National Institute for Materials Science, Tsukuba, Ibaraki, Japan. [4] ICREA-Institució Catalana de Recerca i Estudis Avançats, Barcelona, Spain. ✉email: roshan.krishnakumar@icfo.eu; frank.koppens@icfo.eu

The photoresponse of semiconductor heterostructures is at the heart of modern optoelectronics[1,2]. A general prerequisite for photo-induced currents is the lack of an inversion centre, whether it is extrinsically defined by doping inhomogeneities in the form of PN junctions, or due to a broken inversion symmetry in the crystal structure. In this regard, the structures of moiré superlattices[3–7] are well suited for photoresponse applications, since the crystal symmetry can be easily reduced by a twist-angle ($\theta$)-induced atomic-scale reconstruction[8,9]. Minimally twisted bilayer graphene (mTBG, $\theta < 0.1°$) represents such a class of moiré superlattices, in which lattice reconstruction of the twisted bilayer is energetically favourable and generates alternating triangular domains of AB/ BA Bernal-stacked regions separated by narrow domain wall networks[8,10,11] (Fig. 1a). Whereas the AB/BA regions locally are identical to Bernal stacked bilayer graphene, the atomic registry between AB sublattices in each layer changes smoothly through the domain walls, dramatically changing the local electronic properties over length scales of ~10 nm[10]. From the perspective of photoresponse, those sharp changes in electronic spectra can serve as local junctions, thus providing an intrinsic photoactive region created by the moiré superlattice. Although these domain wall networks have already been shown to strongly influence the DC transport[12–15] and optical properties[16–23] of mTBG, their optoelectronic properties have not yet been explored. In part, this is challenging due to the sub-diffraction scale of the moiré structure that makes it difficult to resolve superlattice scale features in typical far-field photoresponse experiments[24,25].

In this work, we perform near-field photocurrent nanoscopy in mTBG. With this, we probe nanoscale photocurrents in mTBG and observe a unique photo-thermoelectric effect[26–29] governed by the symmetry breaking of the domain wall network. Our measurements are supported by simulations of the spatial photocurrent profile in our devices that shows the photoresponse originates from microscopic variations in the Seebeck coefficient intrinsic to the moiré lattice. By varying the doping, we observe anomalous sign reversals in the photocurrent that hints to the importance of strain gradients in mTBG. In addition, we observe a localised photoresponse close to the domain walls of the moiré unit cell that is attributed to additional electron heating from propagating polaritons.

## Results

**Experimental techniques and device characterisation.** To study the photoresponse of our samples, we employed infrared scanning near-field photocurrent microscopy[30–32]. The technique involves local photoexcitation of carriers using a scattering-type scanning near-field optical microscope[33] (s-SNOM) combined with an electrical current read out at one of the device contacts (see "Methods" section). For local photoexcitation, we focus infrared light (with photon energy ~100–200 meV) onto a sharp tip (Fig. 1a), inducing a strongly localised electric field at the tip apex in an area similar to the tip radius (20–30 nm). By scanning the tip around our sample, we build spatial maps of the photocurrent generated by this localised field. We note that in some cases we measured photovoltage, which is linearly related to the photocurrent through the device resistance $R$ as $I_{PC} = V_{PV}/R$. In this way, we can measure the photoresponse between two pairs of contact simultaneously. On the same sample, we measured the near-field optically scattered light from the tip using interferometric detection[34] (see "Methods" section). This provides a complementary characterisation of the local optical response in our samples.

The studied mTBG samples were fabricated using the tear and stack method[7,35] (see "Methods" section). In brief, the heterostructure structure consists of a Van der Waals stack of mTBG fully encapsulated with hexagonal boron-nitride (hBN) and deposited on top of a Si/SiO$_2$ wafer[36]. A sample (Fig. 1b) also contains gold contacts for photocurrent/photovoltage read out and simultaneous electrostatic gating of the channel with the silicon backgate.

To characterise the local structure in our devices, we first measured the near-field optical scattering signal of our samples. Figure 1c shows a near-field image of the optically scattered light measured at high doping $n \sim 5 \times 10^{12}$ cm$^{-2}$ in our mTBG sample (see Supplementary Note 1 for doping estimation). It reveals a set of bright fringes forming a triangular network of alternating AB/ BA domains; the purple/yellow triangles highlight two neighbouring AB/BA domains. These features have already been studied extensively in near-field scattering experiments and were attributed to enhanced optical conductivity at the domain walls[20–22] of mTBG. Their observation in our experiment thus confirms the presence of atomic reconstruction[8] expected in

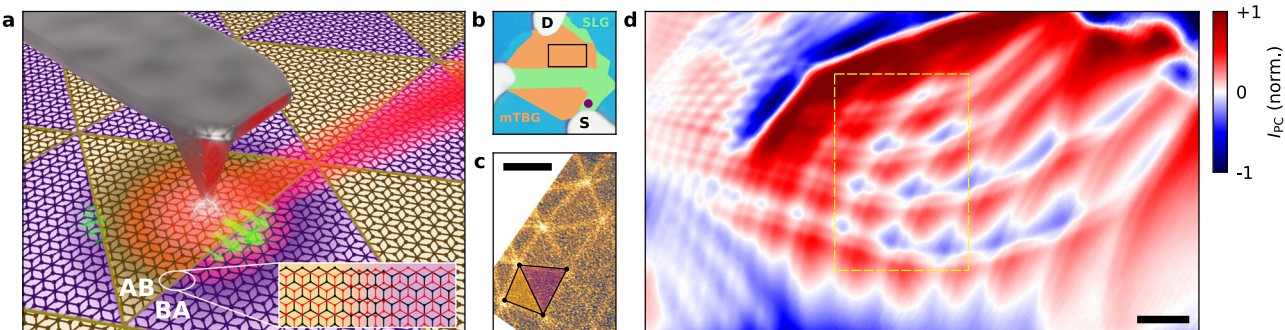

**Fig. 1 Near-field photoresponse in minimally twisted bilayer graphene. a** Schematic illustration of near-field photocurrent experiments performed in minimally twisted bilayer graphene. The moiré domains of different stacking configurations are highlighted by yellow and purple shaded areas. The AFM tip couples infrared light into the device, causing the electron temperature to elevate locally. The photo-thermoelectric effect converts this heat partially into a current that can be read out by the contacts. Inset: Zoom of the domain wall structure separating AB-BA domains of the more lattice. **b** Device schematic of the main device under study. It consists of different regions of single-layer graphene (SLG) and minimally twisted bilayer graphene (mTBG) due to folding/ stacking faults during the heterostructure assembly. The geometry for photocurrent measurements in **d** is marked by S (source) and D (drain). The third contact is left floating. The purple dot marks the position where the calibration of carrier density is done (Supplementary Note 1). **c** Near-field scattering phase image corresponding to the yellow dashed rectangle in **d**, measured at $E = 117$ meV and $n \sim 5 \times 10^{12}$ cm$^{-2}$. The two shaded triangles mark two domains. Length of scale bar: 500 nm. **d** Photocurrent map of mTBG corresponding to the black rectangle in **b**, measured at carrier density $n \sim 1 \times 10^{12}$ cm$^{-2}$ with an excitation energy $E = 188$ meV. Length of scale bar: 500 nm. The map is normalised by the maximum measured $I_{PC}$.

mTBG and allows direct structural mapping of the moiré lattice in our samples.

**Photoresponse in mTBG.** Figure 1d shows a near-field photocurrent map of our sample measured in the mTBG region with an excitation energy $E = 188$ meV, which exhibits a number of interesting features. First, a clear periodicity in the photocurrent is observed throughout the entire sample. It can be easily seen by following the zeros in photocurrent (white lines and white features) that trace the periodicity intrinsic to mTBG. Notably, the periodicity varies from ~100 to ~1000 nm in different regions of the device. We attribute this behaviour to local variations in the twist angle ($\theta \sim 0.1$–$0.01°$) inherent to twisted moire superlattices[37,38]. Second, the photocurrent exhibits alternating domains of a negative response (blue regions) and positive response (red regions). This is most clearly seen in the device region with the largest moiré periodicity (centre area in Fig. 1d). Third, in these larger areas, a second fringe can be seen running parallel to the domain walls. This double step-like feature closely resembles that of propagating polaritons typically observed in near-field scattering and photocurrent experiments on graphene[31,39,40] and hBN[41,42] close to crystal edges, and will be discussed in detail below. Importantly, this periodic structure in the photocurrent was characteristic in our other studied mTBG devices (Supplementary Note 2).

The spatial pattern of photoresponse is highly sensitive to the position of measurement contacts with respect to the moiré periodicity. Figure 2a shows a zoomed photovoltage map of the region with the largest domains in Fig. 1d. This triangular pattern resembles the moiré pattern of mTBG (Fig. 1a), with zero crossings seemingly tracing the domain wall network of the moiré superlattice. However, when comparing our photoresponse data with our optical scattering data (Fig. 1c), we find that the actual structure of the moiré lattice is rather different. The purple/yellow shaded triangles in Fig. 2a highlight the positions and orientation of two neighbouring AB/BA domains measured by optical scattering (see corresponding triangles in Fig. 1c). From this, we can see that this measurement does not capture the entire structure of the moiré lattice, and we lose a set of domain walls at ~45° to those observed in Fig. 2a. Strikingly, those additional domain walls appear simply by measuring between a different pair of contacts (Fig. 2b). In this scheme, the actual structure of the moiré lattice is clearly visible and the spatial profile of the photoresponse is more complex than anticipated. For example, we find that the measured photovoltage exhibits sign reversals not only at domain wall boundaries (Fig. 2c) but also within the AB/BA stacked domains. Moreover, the spatial patterns of the photoresponse within each moiré unit cell varies significantly between AB and BA stacking configurations.

To elucidate the role and mechanism of photoresponse in mTBG, we studied the gate-voltage dependence of the photocurrent within the moiré domains. In Fig. 2d, we plot the line trace of the measured photocurrent made across domain walls (see black dashed line in Fig. 2a) for different gate voltages $V_G - V_D$, where $V_D$ is the position of the charge neutrality point (CNP) determined from the gate dependence of the optically scattered signal (Supplementary Note 3). The positions of domain walls are marked by the black arrows. The first thing to note is that the gate-voltage dependence of the photoresponse is sensitive to the position of the excitation spot within the moiré unit cell. On top of that, it is non-monotonic, such that for some excitation positions it exhibits up to three sign changes. Figure 2e plots line cuts of the gate-voltage dependence for two excitation positions (marked by coloured dashed lines in Fig. 2d). On one hand, the

observed non-monotonic behaviour strongly resembles that of the photo-thermoelectric effect in graphene[26,27,29] that is strongest for lower $n$ and exhibits sign reversals around CNP. This is not surprising considering the unit cell of mTBG is comprised mostly of AB Bernal stacked bilayer graphene, whose photoresponse is dominated by the photo-thermoelectric effect in the presence of spatially varying Seebeck coefficients[26,27,29]. On the other hand, some other features such as the two additional sign changes away from charge neutrality, which depend on the spatial location, are rather peculiar.

**Photo-thermoelectric effect.** With Fig. 2 in mind, we constructed a model based on the photo-thermoelectric effect (PTE) to describe the observed photocurrent features in our device. Photocurrent generation due to the PTE proceeds in four steps. First, the electric field generated by the tip induces an oscillating current density in the sample at the excitation frequency $\omega_{ph}$. Second, this oscillating current causes Joule heating of the electron gas. The power density produced is proportional to the square of the current and has a rectified component $Q(\mathbf{r}, \mathbf{r}_{tip})$,

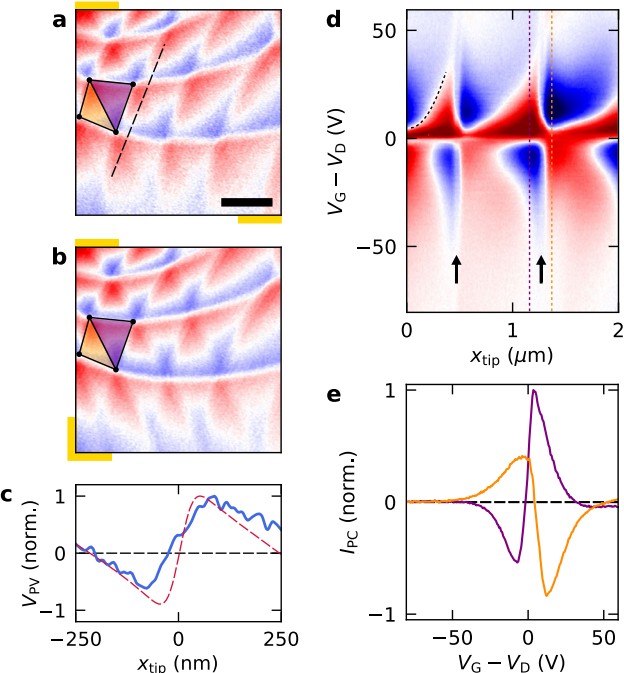

**Fig. 2 Photoresponse in the moiré unit cell of mTBG. a** Zoomed map of measured photovoltage $V_{PV}$ in the same sample presented in Fig. 1 (corresponding to the area with the largest domains in Fig. 1d), measured at excitation energy $E = 117$ meV and carrier density $n \sim 1 \times 10^{12}$ cm$^{-2}$. The map is normalised by the maximum measured $V_{PV}$. Colour code: blue: $-1$, red: $+1$. The yellow/purple shaded triangles correspond to those in Fig. 1c. The gold annotations illustrate roughly the relative position of contacts used for measuring voltages. Length of scale bar: 500 nm. **b** Same as **a** but for a different choice of contacts (gold annotations) measured simultaneously. **c** Line-cut across a horizontal domain wall in **b** (blue), together with a line trace of the simulated profile in Fig. 3g (dashed red). **d** Photocurrent as a function of gate-voltage $V_G$ (with respect to the position of the Dirac point $V_D$) measured for a line trace that crosses several domain walls (see black dashed line in **a**). Black arrows mark the position of observable domain walls for this choice of contacts. The gate voltages correspond to carrier densities within roughly $\pm 6 \times 10^{12}$ cm$^{-2}$. **e** Gate-voltage response along the two line-cuts highlighted in **d**, on two opposite sides of the domain wall.

where $\mathbf{r}$ is the position and $\mathbf{r}_{\text{tip}}$ the position of the tip. Third, the generated heat spreads in the sample on a characteristic length scale referred to as the cooling length[28] ($L_{\text{cool}}$) before being dissipated to the substrate. Finally, since the heat transport is coupled to the charge transport via the Seebeck–Peltier effect, the heat flux is able to generate a net electric current in the presence of Seebeck coefficient gradients. The current and heat flux are governed (at least for small incident power) by a set of linear equations with respect to the source term $Q(\mathbf{r}, \mathbf{r}_{\text{tip}})$. As a consequence, the PTE-induced photovoltage $V_{\text{PTE}}^{(m)}$ at the contact $m$, with respect to a grounded contact used as a common reference, is then given (see details in Supplementary Note 4) by a linear relation of the type

$$V_{\text{PTE}}^{(m)} = \int d\mathbf{r}\, \mathcal{R}_{\text{PTE}}^{(m)}(\mathbf{r}) Q(\mathbf{r}, \mathbf{r}_{\text{tip}}), \tag{1}$$

where $\mathcal{R}_{\text{PTE}}^{(m)}(\mathbf{r})$ is the photovoltage responsivity function that encodes the PTE response of the system. An analogous formula holds for photocurrents. In absence of strong resonant features, for example plasmonic excitations, the response of the system at the energy $\hbar\omega_{\text{ph}}$, and therefore $Q(\mathbf{r}, \mathbf{r}_{\text{tip}})$, rapidly decays with $|\mathbf{r} - \mathbf{r}_{\text{tip}}|$. This means that $V_{\text{PTE}}^{(m)} \propto \mathcal{R}_{\text{PTE}}^{(m)}(\mathbf{r}_{\text{tip}})$ and the photovoltage (or photocurrent) maps are essentially measuring the responsivity (apart from the convolution with a spread function due to the finite tip dimension).

To gain some intuition into the origin of the spatial profile of the photoresponse generated in our devices, we consider the form of the photovoltage responsivity function for a one-dimensional device (see derivation in Supplementary Note 5). This reads

$$\mathcal{R}(x) = -\frac{1}{\kappa W}\int dx'\, \frac{L_{\text{cool}}}{2} e^{-\frac{|x'-x|}{L_{\text{cool}}}} \partial_x S(x'), \tag{2}$$

where $W$ is the width of the device and $\kappa$ its electronic thermal conductivity. In this example (in open-circuit configuration), an infinitely sharp tip creates a spatial profile of electron temperature (Fig. 3a) in the form $\delta T(x) \equiv T(x) - T_0 \propto e^{-|x-x_{\text{tip}}|/L_{\text{cool}}}$, with $T_0$ being the temperature of the substrate. From (2), we clearly see that the PTE requires an inhomogeneous Seebeck coefficient $S$ with a gradient parallel to the path between measurement contacts. In addition to that, the responsivity decays away from the Seebeck coefficient fluctuations on a length scale $L_{\text{cool}}$, since that is the distance that heat is able to travel in the sample. Finally, in a system with particle-hole symmetry, $\mathcal{R}_{\text{PTE}}(x)$ is an odd function of the electronic density because $S(x)$ is odd. This behaviour is clearly observed in the gate-voltage dependence of the measured photoresponse (Fig. 2e), where the photoresponse changes sign around the Dirac point due to a change in sign of the carriers' charge.

In the moiré superlattice of mTBG, the local layer alignment transitions smoothly in the region of the domain walls, which is expected to cause local gradients in the Seebeck coefficient, as required for the PTE. To corroborate this, we calculated (see Supplementary Note 6 and "Methods" section for full details on the calculation) the Seebeck coefficient of bilayer graphene as a function of the layer alignment in the Relaxation Time Approximation[43] and mapped the local alignment to the distance from a domain wall using the result in ref. [10]. By this procedure, we obtained the spatial profile of the Seebeck coefficient across a domain wall, as depicted in Fig. 3b. This shows that the Seebeck coefficient dips sharply at the domain walls of the moiré superlattice. By considering Eq. (2) and recalling that $V_{\text{PTE}}^{(m)} \propto \mathcal{R}_{\text{PTE}}^{(m)}$, we plot the expected photovoltage profile in the vicinity of the domain walls (Fig. 3c). Notably, the photovoltage follows a non-monotonic dependence with the

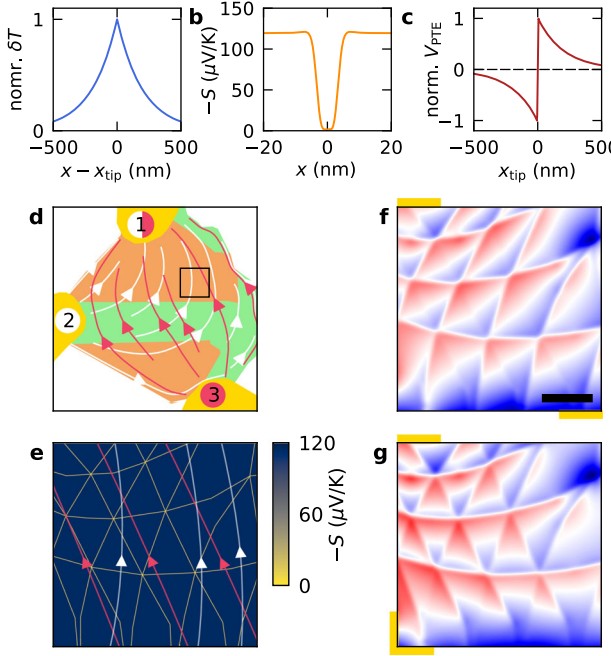

**Fig. 3 Simulations of the photo-thermoelectric response in mTBG. a** Profile of the electron temperature increase induced by local photoexcitation as a function of distance ($x$) from the excitation position ($x_{\text{tip}}$), for a cooling length of 200 nm. **b** Calculated Seebeck coefficient as a function of position within two moiré unit cells separated by a domain wall at $x = 0$ nm. **c** Line trace of the calculated photovoltage across a domain wall using the thermal profile and Seebeck coefficient of **a** and **b**. **d** Illustration of the geometry used in our photocurrent simulations. The yellow semi-circles indicate contact probes, the green single-layer graphene regions and the orange mTBG regions. The red projection of current flows corresponds to a configuration wherein current flows between contacts 3 and 1, while the white field lines correspond to the current flowing between contacts 2 and 1. The black square highlights the region shown in **e**–**g**. **e** 2D spatial map of the calculated Seebeck coefficient (Supplementary Note 6) that goes into our model, along with the projections of current flows shown in **d**. **f** Zoom of the photovoltage simulations in the region of mTBG (same area as Fig. 2a). Here we convolute the calculated responsivity with a Gaussian function of width 15 nm to account for the finite tip radius. Colour code: blue: −1, red: +1. Length of scale bar: 500 nm. **g** Same as in **f** but for a different measurement geometry (same as Fig. 2b).

position through the domain wall, changing sign as it crosses the middle of the domain wall. Such behaviour can indeed be seen in our data along certain domain walls (Fig. 2c).

Although the one-dimensional model (Fig. 3c) describes well the sign changes across certain domain walls (Fig. 2c), the full two-dimensional spatial map can be more complex as one must consider the photocurrent contributions from different domain walls due to the finite cooling length $L_{\text{cool}}$. The full spatial profile of $\mathcal{R}_{\text{PTE}}^{(m)}(\mathbf{r})$ can be calculated if the entire measurement geometry of the system (Fig. 3d) and the spatial profile of the Seebeck coefficient (Fig. 3e) are known. We perform this calculation using the Finite Element Method (see Supplementary Note 4 and "Methods" section for details). The calculation is greatly simplified thanks to an elegant reciprocity relation[44,45].

Figure 3f, g plots the simulated photocurrent for the two measured contact configurations (Fig. 3d) in the same area corresponding to the measurement in Fig. 2a and b, respectively. The simulations are carried out at a fixed doping $n = 1 \times 10^{12}\ \text{cm}^{-2}$. Comparing Fig. 3f, g with Fig. 2a, b, we find good agreement between our measurements

and the simulations. Not only do our simulations accurately capture the spatial sign changes observed in our sample, but also the differing local photoresponse between AB and BA stacked regions (Figs. 2b and 3g). Importantly, the simulations also capture the strong directional effect observed. To understand this behaviour, we recall that the PTE can generate a global current only in regions where the gradients of the Seebeck coefficient contain a part running parallel to the projection of current flows between contacts. This means that the domain walls perpendicular to the projection of current flows contribute strongest to the measured photocurrent and are minimum for those that run parallel. The current projections for the two contact configurations are sketched in Fig. 3d, where the red and white lines depict projections for the contact configuration of Fig. 2a and b, respectively. They are also drawn in Fig. 3e to illustrate their orientation relative to the domain wall networks measured in Fig. 2a, b. In the first configuration (red lines) we find that some domain walls run almost parallel to the current projections. Therefore, photoexcitation at these domain walls does not contribute to the globally measured current, and they remain completely hidden in our measurement (Fig. 2a). Whereas in the second case (white lines), there is always a component perpendicular to all sets of domain walls such that they all contribute to the globally measured current (Fig. 2b). This result also demonstrates the crucial importance of measurement geometry[46] in understanding the nanoscale photo-response of solid-state crystals.

Good agreement between the simulations and experiment points towards a photo-thermoelectric dominated photoresponse in the moiré lattice of mTBG. Indeed, qualitatively the spatial sign changes across domain walls and doping dependent sign changes around the CNP are well described by our photo-thermoelectric model; we attribute the slight shift of zero crossings from $V_G - V_D = 0$ (Fig. 2d, e) to additional photocurrent mechanisms that may play a role at the CNP[47]. Even the additional sign changes observed away from CNP (see the purple trace in Fig. 2e), can be explained within the framework of the PTE by considering an enhanced Seebeck coefficient at the domain walls with respect to the calculated one (Supplementary Note 7). That said, it does not accurately describe the spatial dependence of the gate-voltage response, specifically, the sign reversal of the photoresponse within the moiré domains shifting with applied gate-voltage (dashed line in Fig. 2d). This behaviour points towards a spatially varying parameter not considered in our model, for example, a spatially varying Seebeck coefficient within the AB/BA domains themselves that is not localised at the domain walls. Such behaviour might be expected in the case that lattice reconstruction in mTBG imposes significant strain in the AB regions[48], which might enhance the Seebeck coefficient of bilayer graphene locally[49]. A simplified model, which includes a spatially varying Seebeck within the AB/BA domains, is presented in Supplementary Note 7 and shows similar features as the experimental data.

Another peculiarity in the data is the high doping behaviour ($|V_G - V_D| > 40$ V), in which the spatial sign changes across domain walls become nearly absent (Fig. 2d) and the spatial profile resembles that of a constant background, which changes sign when changing carrier polarity. This is illustrated in Fig. 4a that plots a photocurrent map for a doping $n \sim 4 \times 10^{12}$ cm$^{-2}$, where we observe large areas of the moiré lattice exhibiting either a constant positive or negative photoresponse. In the simplest case, we might attribute this additional photocurrent to nearby PN junctions caused by deformations/stacking faults in our heterostructure. However, our measurements of the cooling length from these interfaces ($L_{cool} = 240$ nm) show a

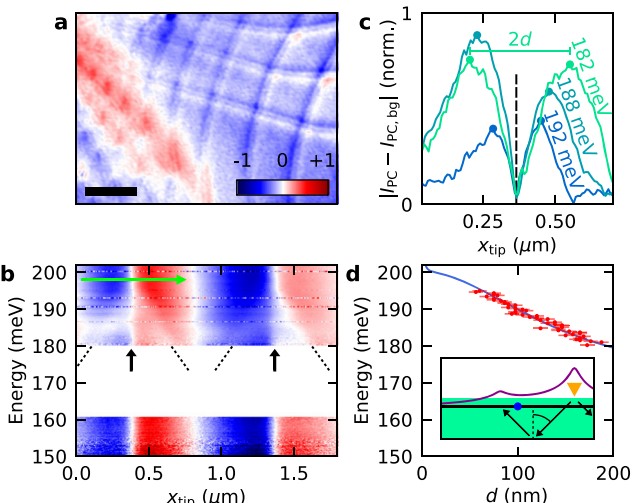

**Fig. 4 Anomalous photocurrent and non-local heating via polariton rays.**
**a** Spatial map of the normalised photocurrent ($I_{PC}$) at an excitation energy of 117 meV and doping $n \sim 4 \times 10^{12}$ cm$^{-2}$. Length of scale bar: 500 nm.
**b** Normalised $I_{PC}$ as a function of excitation energy and excitation position ($x_{tip}$) crossing two domain walls (marked by black arrows) in another of our mTBG samples. Colour code: blue: $-1$, red: $+1$. The dashed lines are guides to the eye showing the dispersive nature of the double-step feature, which appears as a local reduction of the photocurrent magnitude on either side of the domain wall. The white area corresponds to a gap in the spectrum of our laser. **c** Line traces (as absolute value) of a few energies taken from **b** (see green arrow in **b**) plotted after background subtraction; the background is experimentally extracted from the line trace at 202 meV where the double-step feature is not observed. We define the distance between the peaks as $2d$. **d** Distance between the domain wall and observed peak ($d$) plotted for different excitation energies (red dots), along with a fit to $a\lambda_{ray}$ (blue line). We estimate the error in $d$ to be ±7.5 nm based on the spatial resolution of our scan. The inset is a schematic showing the thermal profile (purple) induced by light coupled in via the AFM tip (yellow triangle). In addition to heating the electron lattice, the tip also excites hBN phonon rays (black arrows), which are able to propagate to the opposite side of the domain wall (blue dot) and cause additional heating that produces a photocurrent contribution of the opposite sign.

fast decay of such contributions (Supplementary Note 8). Hence, the data suggests another photocurrent mechanism might be present in mTBG. For example, we considered the possibility of photogalvanic currents (Supplementary Note 9). Further work is required to understand these additional background phenomena.

**Photoresponse from hyperbolic phonon–polaritons.** Finally, we address the double step-like feature that is observed close to the domain walls (Figs. 1d, 2d and 4a). At first inspection, those features resemble that of propagating polaritons in graphene/hBN heterostructures typically observed in s-SNOM experiments[39–41,50,51]. As shown in this work, the domain walls also act as local photoactive junctions which would thus enable thermoelectric detection of polariton modes[42,52]. For the investigation, we studied the wavelength dependence of the double-step feature focussing on energies 150–200 meV where it appeared strongest (Fig. 1d). Figure 4b plots the photo-response as a function of excitation wavelength for a line trace made across several domain walls in one of our mTBG samples. For all energies, we observe the expected photocurrent profile

generated at domain walls by the photo-thermoelectric effect (Fig. 3c). However, we also observe an additional feature that disperses with energy in the specific range 180–200 meV but is completely absent for lower energies. This spectral range corresponds to the upper reststrahlen band of hBN and suggests those features originate from propagating phonon–polaritons in hBN.

For further analysis, we compare in Fig. 4c line traces of the measured photoresponse (green arrow in Fig. 4b) at different energies after subtraction of a smooth background. Note, whereas we plot the modulus to make analysis easier, we draw attention to the fact that the actual response is a reduction in the measured photocurrent compared to the background (Fig. 4b). Following this procedure for all traces in Fig. 4b, we plot in Fig. 4d the distance between peaks and the domain wall as a function of excitation energy (red dots). When the excitation energy is inside the reststrahlen bands of hBN, the s-SNOM tip can excite hyperbolic phonon–polaritons that travel in hBN as collimated rays that propagate with a fixed angle $\theta_{BN} = \tan^{-1}\left[\text{Re}\left(\frac{i\sqrt{\in_{x,y}}}{\sqrt{\in_z}}\right)\right]$ relative to the vertical direction[53] ($\in_i$ being the components of the dielectric function of hBN). This produces a series of maxima of electric field intensity close to the surface separated by a distance $\lambda_{ray} = 2t\cdot\tan\theta_{BN}$, where $t$ is the total thickness of the hBN. We note our photocurrent measurements are more sensitive to ray-like modes rather than the first eigenmode typically observed in s-SNOM experiments[41,54,55] (see details in Supplementary Note 10). Following this insight, we fit the distance $d$ between the dispersing feature and the DW (Fig. 4d) with the function $d(\omega) = a\lambda_{ray}(\omega)$, with $a$ as a proportionality constant. The fit yields $a = 0.63$ and shows good agreement with experimental data (Fig. 4d) providing strong evidence that the dispersing feature is caused by hBN hyperbolic phonon–polaritons. We observe that $a < \frac{t_{top}+2t_{bot}}{2t_{top}+2t_{bot}} = 0.95$ (hBN ray passing once through the top-hBN and twice through the bottom hBN), which suggests that phonon-polariton rays launched by the s-SNOM tip propagate beyond the DW into neighbouring domains. This is consistent with the fact that the dispersing feature traces a reduction in the measured photocurrent because propagating phonon–polaritons heat the mTBG on the side of the domain wall opposite to the tip (sketched in the inset of Fig. 4d). This produces photocurrents that counteracts the one generated by the heat directly produced by the tip, leading to a small reduction of the photocurrent signal (Fig. 4b). We note that Figs. 2d and 4a also show a fainter double-step feature, while the data are taken outside the reststrahlen bands of hBN. We speculate that the double-step feature, in this case, might originate from heating by propagating plasmon polaritons that have been shown to scatter/reflect from domain walls of mTBG[20–22].

## Discussion

To conclude, we show that near-field photocurrent spectroscopy is a valuable tool for studying the optoelectronic properties of moiré superlattices. Our moiré-scale resolved measurements reveal a spatially rich photoresponse governed by the symmetries of the reconstructed lattice that would go unnoticed in typical far-field photoresponse experiments. Good agreement of our simulations with experimental data shows the importance of hot carriers in the photoresponse of mTBG and, at the same time, shows the crucial link between global measurements and local excitation in photocurrent experiments. Our work should thus motivate further near-field photocurrent studies on related moiré superlattices including twisted transition metal dichalcogenides[9] and small-angle twisted bilayer graphene. In the

course of preparing this manuscript, we became aware of related work[56].

## Methods

**Device fabrication**. The mTBG device discussed in the main text was fabricated using a standard tear-and-stack method[7,35] with a set twist angle of 0.15°. Here we use a polycarbonate film to pick up the individual flakes to form an hBN (30 nm)/mTBG/hBN(6 nm), which is released at 180 °C on a Si/SiO$_2$(285 nm) substrate. 1D contacts are patterned by photolithography, followed by Cr(5 nm)/Au(50 nm) deposition[36]. To avoid picking up contamination during our measurements, we clean the surface from polymers residues by AFM-brooming[57]. Other samples were fabricated in a similar manner. The data discussed in Fig. 4b–d are taken in an hBN(75 nm)/mTBG/hBN(8 nm) stack.

**Measurement details**. We used an s-SNOM from Neaspec that allows simultaneous measurement of the scattered near-field signal and any photocurrent/photovoltage signals. We used a CO$_2$ laser (Access Laser) and a tunable QCL laser (Daylight Solutions) as infrared light sources between 110 and 250 meV (5–11 μm), at a typical power between 20 and 40 mW. This light is focused on a PtIr-coated AFM tip (Nanoworld), which is oscillating above the sample surface at ~250 kHz with a tapping amplitude of 80–100 nm. The modulated scattering signal is detected by a fast cryogenic HgCdTe detector (Kolmar Technologies), and by operating the s-SNOM in pseudoheterodyne mode we can independently measure the scattering amplitude and phase[34]. For photocurrent measurements, we used a fast current amplifier (Femto DLPCA-100). For simultaneous measurement of the photovoltage between two pairs of contacts, we used two differential voltage amplifiers (Ithaco 1201) with one common contact grounded. The carrier doping in our samples is tuned by applying a DC voltage between the Si backgate and our device. To avoid detecting unwanted far-field contributions to the scattered or photocurrent/voltage signal, we detect the near-field signals at the 2nd or 3rd harmonic of the cantilever oscillation.

The measured photocurrent/voltage signal is demodulated with the driving signal of the AFM cantilever as a reference signal. However, the actual motion of the AFM cantilever can have a phase offset that varies with the position on the sample (due to tip–sample interaction). This phase offset is given at each pixel by the measured phase delay between the tip driving signal and the actually detected motion. To correct our photocurrent/voltage signal measured at harmonic $i$, we subtract at every point $i$ times this phase delay. In addition to this, there remains a global phase offset in the corrected photocurrent/voltage signal due to the electronics in the circuit. Since the photocurrent/voltage signal is a real-valued quantity, we subtract this global phase offset, which we determine by taking the most frequent phase within a scan.

**Numerical simulations**. Finite element method (FEM) simulations of the PTE response of the device were performed using an open-source (LGPLv3), home-made, python package (available at https://gitlab.com/itorre/diffusive_solver)[44] based on the FEniCS library[58], that allows the solution of coupled diffusion equations systems in realistic sample geometries. Band-structure calculations of shifted graphene bilayers and the extraction of the corresponding physical properties were carried out using an open-source (LGPLv3) python package (available at https://gitlab.com/itorre/bandstructure-calculation), which allows the computation of the spectrum and of optical and thermoelectric properties of simple electronic band-structure models.

## Data availability

The data that support the plots within this paper and other findings of this study are available from the corresponding authors upon reasonable request.

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

## Acknowledgements

We thank Justin Song for fruitful discussions and are grateful to Matteo Ceccanti for providing the illustration in Fig. 1a. The research leading to these results has received funding from the European Union's Horizon 2020 research and innovation programme under grant agreement Ref. 881603 (Graphene Flagship Core Project 3). Furthermore, this work was supported by the ERC TOPONANOP under grant agreement Ref. 726001. N.C.H.H. also acknowledges funding from the European Union's Horizon 2020 programme under the Marie Skłodowska-Curie grant agreement Ref. 665884. I.T. also acknowledges funding from the Spanish Ministry of Science, Innovation and Universities (MCIU) and State Research Agency (AEI) via the Juan de la Cierva fellowship Ref. FJC2018-037098-I. D.B.R. also acknowledges funding from the Secretaria d'Universitats i Recerca del Departament d'Empresa i Coneixement de la Generalitat de Catalunya, as well as from the European Social Fund. H.H.S. also acknowledges funding from the European Union's Horizon 2020 programme under the Marie Skłodowska-Curie grant agreement Ref. 843830. K.W. and T.T. also acknowledge support from Japan's MEXT under the Elemental Strategy Initiative (grant Ref. JPMXP0112101001), from a KAKENHI grant by the JSPS (Ref. JP20H00354), and from the JST under the CREST programme (grant Ref. JPMJCR15F3). R.K.K. also acknowledges funding from the European Union's Horizon 2020 programme under the Marie Skłodowska-Curie grant agreements with Ref. 754510 and Ref. 893030. F.H.L.K. also acknowledges support from the Government of Spain (FIS2016-81044; Severo Ochoa CEX2019-000910-S), Fundació Cellex, Fundació Mir-Puig, and Generalitat de Catalunya (CERCA, AGAUR, SGR 1656). F.H.L.K. also acknowledges support from the PID2019-106875GB-I00 project funded by MCIN/ AEI /10.13039/501100011033.

## Author contributions

F.H.L.K. and N.C.H.H. conceived the experiment. N.C.H.H. and R.K.K. conducted the experiments, with assistance from D.B.R. and H.H.S. Devices were fabricated by N.C.H.H. and R.K.K., with K.W. and T.T. providing hBN crystals. N.C.H.H., I.T., R.K.K. and F.H.L.K. analysed and interpreted the results. N.C.H.H. and I.T. performed numerical simulations using the band-structure calculator and FEM solver developed by I.T. The manuscript was written by R.K.K., I.T., N.C.H.H. and F.H.L.K. with input from all authors. F.H.L.K. supervised the work.

## Competing interests

The authors declare no competing interests.
