## [Peer Review File · Nature Communications]

REVIEWER COMMENTS

Reviewer #1 (Remarks to the Author):

By using the scanning probe photocurrent microscopy, Hesp et al. demonstrated rich details of photocurrent response in graphene moiré superlattice. This work shows that scanning photocurrent microscopy is useful for studying the optoelectric properties in moiré superlattice structures at nanoscale length. The imaged photocurrent grid pattern is due to the Seebeck coefficient mutation at the twisted graphene domain wall. Through controlling the carrier gas's concentration, the sign and magnitude of the photoresponse can be effectively tuned. The photoresponse pattern is sensitive to the domain wall's orientation with respect to measurement contacts. This work displays a novel route to probe the optoelectronic response in moiré superlattice and is of interest to others in the community and the broader field. The calculated and simulated results agree with the experimental observation. This work is an excellent representative of probing the photoelectric response in moiré superlattice with nanoscale resolution.

I recommend this manuscript to be accepted after considering the following comments.

Comment :

1. In figure 4c, the author attributes the two side peaks to the interference between back-reflected hBN phonon polariton and the incident light field. Considering the phonon polariton is a bulk mode. The two side peaks should be attributed to the phonon polariton launched by domain walls. Besides, the dispersion calculation in figure 4d deviates from the experimental results (solid red circles); why?

2. In Fig. 2d and 2e., the authors point out that the photocurrent within moiré domains exhibits sign reversals at the CNP. What is the offset gate voltage at the CNP? Does the intersection of the curves drawn in Fig. 2e with the $I_{pc}=0$ near $V_G-V_D=0$ represent CNP? If so, why the intersection points are not at 0?

Reviewer #2 (Remarks to the Author):

The manuscript „Nano-imaging photoresponse in a Moire unit cell“ by N.C.H. Hesp et al reports on interesting new results obtained by near-field photocurrent-imaging of minimally twisted Bilayer graphene (mTBLG) on the Moire cell lengthscale. By using a s-SNOM tip for local illumination and collecting the photocurrent of contacted mTBLG sample, the authors observe rich spatially varying patterns, where AB-BA domain boundaries and a strong directional effect are intermixed. They try to explain these rich patterns in detail based on the photo-thermoelectric response of mTBLG and achieve good agreement with their numerical simulations. They also observe additional spatial features in the photocurrent which are linked to the dispersion of phonon-polaritons in the encapsulating hBN layers.

The manuscript shows that the local photocurrent response gives rich information of twisted BLG inside the Moire unit cell. It is definitely a very timely contribution to the exciting field of twisted vdW materials and will stimulate further research and refinement. While the overall agreement of their data with their experiments is very convincing, the complex sample layout and observed patterns make it not easy for the reader to follow. I believe the manuscript could further be improved in terms of readability by adding clearer graphs and more step-by step explanations, especially in the beginning part.

Overall, I can recommend publication of this manuscript in Nature Communications after a minor

revision.

Detailed comments:

1) Figure 1 should be improved in terms of clarity:

- The sketch in panel a) is meant to show the AB/BA domain (walls) and the measurement principle. The different stacking configurations are displayed in yellow and purple, respectively. However, the domain right below the tip (AB) is displayed in purple, although it should be yellow (in order to distinguish it from the neighboring BA domain). I know that this panel a) just acts as a sketch, but such inconsistent coloring scheme confuses the reader unnecessarily.
- In panel b), it is unclear, at which position on the sample the s-SNOM image (panel c) and the local photocurrent (PC) map (panel d) have been taken. I do see two rectangles in the sketch, which could indicate the positions of both s-SNOM and local PC maps, but this information is missing from the caption.
- The comparison of s-SNOM and local PC maps would be much easier for the reader if they were showing the same area and would be of the same size. (e.g. like the Zoom-in of Figure 2a & b).

2) Figure 2 should be as well improved in terms of clarity:

- In the present version of Fig. 2 a & b, the domain walls (yellow and purple triangles) seem to appear from nowhere as I cannot see the correspondence to the s-SNOM image shown in Fig. 1c).
- In the main text, the sentence "The black traces in Figure 2a outline the domain wall network of the moiré superlattice measured by optical scattering. " I assume the authors talk about the outlines of the yellow and purple triangles, which only indicate 2 domains. From the wording, I expected many (or all) domains being traced. Also, the wording "black trace" can be misleading as in the figure caption a line trace is used to extract the PC as function of gate voltage. Maybe a clearer description (or more information in the supporting information) would be helpful to better guide the reader through the manuscript.

Overall, the description in the paragraph (line 90-104) was hard to follow as no direct comparison of s-SNOM (domains) and PC images is given.

3) In line 81, the authors state that "the periodicity varies from 100 nm to 1000 nm in different regions" and attribute this to "local variations of the twist angle". Is this a reasonable assumption? How much would the twist angle need to vary for this? Would one expect a continuous change in periodicity or some abrupt changes? This could be better elaborated.

4) The reference section needs to be checked as many mistakes are present:

Ref. 16: page numbers wrong: should be article number 205404

Ref. 17: page numbers wrong: should be article number 241412(R)

Ref. 18: page numbers wrong: should be article number 161406(R)

Ref. 19: page numbers wrong: should be article number 11760

Ref. 24: article number missing: 10699

Ref. 44: journal name missing, Page number wrong: Should be PHYSICAL REVIEW B 90, 075415 (2014)

Ref. 45: journal name missing, ArXiv or published?

Ref. 48: page range useless, article number sufficient

Ref 52: page numbers wrong: should be article number 073110

Reviewer #1 (Remarks to the Author):

By using the scanning probe photocurrent microscopy, Hesp et al. demonstrated rich details of photocurrent response in graphene moiré superlattice. This work shows that scanning photocurrent microscopy is useful for studying the optoelectric properties in moiré superlattice structures at nanoscale length. The imaged photocurrent grid pattern is due to the Seebeck coefficient mutation at the twisted graphene domain wall. Through controlling the carrier gas's concentration, the sign and magnitude of the photoresponse can be effectively tuned. The photoresponse pattern is sensitive to the domain wall's orientation with respect to measurement contacts. This work displays a novel route to probe the optoelectronic response in moire superlattice and is of interest to others in the community and the broader field. The calculated and simulated results agree with the experimental observation. This work is an excellent representative of probing the photoelectric response in moiré superlattice with nanoscale resolution.

I recommend this manuscript to be accepted after considering the following comments.

Comment:

1. In figure 4c, the author attributes the two side peaks to the interference between back-reflected hBN phonon polariton and the incident light field. Considering the phonon polariton is a bulk mode. The two side peaks should be attributed to the phonon polariton launched by domain walls. Besides, the dispersion calculation in figure 4d deviates from the experimental results (solid red circles); why? We thank the Reviewer for his questions. Despite the close match of the experimental data with the calculations, the small deviation towards higher energy directed us to improve the simple model accounting for the apparent step-like features near the domain walls. We followed the Reviewers suggestion and now consider the phonon polaritons as ray-like modes propagating in the bulk formed by the hBN slabs. This leads to a simpler and better-matching explanation that we detail in the following.

The spectrum of possible hyperbolic phonon-polariton (HPP) eigenmodes in an hBN slab is given by momenta $k_0 + n\Delta k$, with k_0 representing the zeroth order mode typically observed in s-SNOM experiments, followed by equidistant modes at separated by Δk (n is an integer). The s-SNOM tip launches in general a combination of these modes (like a ray). Ignoring losses and considering for simplicity a 1D problem ($q_y = 0$) this combination can be written in a Bloch-wave form as $e^{ik_0(x-x_{\text{tip}})}u(x-x_{\text{tip}})$, where u is a periodic function with periodicity $\lambda_{\text{ray}} \equiv 2\pi/\Delta k$. The question that arises is whether it is more correct to compare the distance d with $2\pi/k_0$ (representing the fundamental eigenmode) or with $2\pi/\Delta k$ (representing the ray-like mode).

In our previous interpretation we compared, in analogy to what is done in the case of HPPs scattering at edges, with $\frac{2\pi}{k_0}$. This is certainly correct when measuring an interference pattern since the phase

$e^{ik_0(x-x_{\text{tip}})}$ is the most important factor in determining the interference. However, our photocurrent generation mechanism is based on the amount of heat that is locally deposited on the sample. This is insensitive to the phase (it is an incoherent mechanism) but sensitive to the intensity pattern encoded in $u(x - x_{\text{tip}})$.

In particular, the periodicity of u creates copies of the field hot-spot generated by the tip at distances multiples of λ_{ray} . When one of these copies (the first one) comes close to the DW on the opposite side, it creates a photocurrent that partially counteracts the one created by the tip hot-spot, leading to a reduction of the signal. λ_{ray} has a nice geometrical interpretation in terms of rays travelling at a fixed angle with respect to the anisotropic axis z of the hBN and can be expressed as $\lambda_{\text{ray}} = 2t \cdot \tan(\theta_{\text{BN}})$ and angle $\theta_{\text{BN}} = \tan^{-1} \left(\text{Re}(i\sqrt{\epsilon_{x,y}} / \sqrt{\epsilon_z}) \right)$ as calculated from the dielectric function ϵ of hBN. With this interpretation we get a better agreement with experimental data, even at the highest energies, and a more sound theoretical understanding.

We have updated the manuscript and the figures according to the new interpretation in the paragraph "Photoresponse from hyperbolic phonon-polaritons", and refer for the details of this analysis to a new Supplementary Section 10.

2. In Fig. 2d and 2e., the authors point out that the photocurrent within moiré domains exhibits sign reversals at the CNP. What is the offset gate voltage at the CNP? Does the intersection of the curves drawn in Fig. 2e with the $I_{\text{pc}}=0$ near $V_{\text{G}}-V_{\text{D}}=0$ represent CNP? If so, why the intersection points are not at 0?

Indeed, $V_{\text{G}}-V_{\text{D}} = 0$ represents the CNP in our samples and is determined from the gate dependence of the optically scattered signal made simultaneously. It essentially measures the Drude response in our samples and changes smoothly with doping, showing a minimum at CNP, which in our case occurs at 11 V. To avoid confusion, we have added this analysis to a new Supplementary Section 3 and added some text referencing this Section (lines 110-113).

The Reviewer is also correct in that the sign reversal in our samples does not always happen exactly at the CNP. We attribute this behaviour to possible additional photocurrent mechanisms that can occur specifically at CNP (see for example Nat. Nanotechnol. 14, 145–150 (2019)), which can contribute a finite photocurrent at the CNP and slightly off-set the photocurrent that is expected. To make this clear, we have included this point in the discussion at lines 202-206 and referenced previous work on photo response at the CNP accordingly.

Reviewer #2 (Remarks to the Author):

The manuscript „Nano-imaging photoresponse in a Moire unit cell“ by N.C.H. Hesp et al reports on interesting new results obtained by near-field photocurrent-imaging of minimally twisted Bilayer graphene (mTBLG) on the Moire cell lengthscale. By using a s-SNOM tip for local illumination and collecting the photocurrent of contacted mTBLG sample, the authors observe rich spatially varying patterns, where AB-BA domain boundaries and a strong directional effect are intermixed. They try to explain these rich patterns in detail based on the photo-thermoelectric response of mTBLG and achieve good agreement with their numerical simulations. They also observe additional spatial features in the photocurrent which are linked to the dispersion of phonon-polaritons in the encapsulating hBN layers.

The manuscript shows that the local photocurrent response gives rich information of twisted BLG inside the Moire unit cell. It is definitely a very timely contribution to the exciting field of twisted vdW materials and will stimulate further research and refinement. While the overall agreement of their data with their experiments is very convincing, the complex sample layout and observed patterns make it not easy for the reader to follow. I believe the manuscript could further be improved in terms of readability by adding clearer graphs and more step-by step explanations, especially in the beginning part.

Overall, I can recommend publication of this manuscript in Nature Communications after a minor revision.

Detailed comments:

1) Figure 1 should be improved in terms of clarity:

- The sketch in panel a) is meant to show the AB/BA domain (walls) and the measurement principle. The different stacking configurations are displayed in yellow and purple, respectively. However, the domain right below the tip (AB) is displayed in purple, although it should be yellow (in order to distinguish it from the neighboring BA domain). I know that this panel a) just acts as a sketch, but such inconsistent coloring scheme confuses the reader unnecessarily.

We agree with the Reviewer and we noticed this was a rendering fault in our image and although the domain is coloured yellow it somehow gives the impression of purple somehow and can be confusing. We have re-edited the image to make the colour contrast more apparent for the domains in the region of the tip.

- In panel b), it is unclear, at which position on the sample the s-SNOM image (panel c) and the local photocurrent (PC) map (panel d) have been taken. I do see two rectangles in the sketch, which could indicate the positions of both s-SNOM and local PC maps, but this information is missing from the caption.

We have clarified this in the new caption of Fig. 1. To make it simpler we only highlight the region in which the photocurrent map was taken (Fig. 1d). The position of the s-SNOM image is then outlined on the Fig. 1d map.

- The comparison of s-SNOM and local PC maps would be much easier for the reader if they were showing the same area and would be of the same size. (e.g. like the Zoom-in of Figure 2a &b).

We agree with the Reviewer that the comparison is slightly confusing. We believe this is because the s-SNOM image of Fig. 1c is rotated compared to the PC maps in Fig. 1D (merely for aesthetic reasons). In the newly modified Fig. 1, we have rotated the s-SNOM image to match the correct orientation of the PC maps and highlighted the region of the PC map where the s-SNOM image was taken (see yellow dashed box on new Fig. 1d). Unfortunately, the range of the s-SNOM scan does not cover the complete photocurrent map, leaving some white gaps. We believe this arrangement makes Fig. 1 clearer.

2) Figure 2 should be as well improved in terms of clarity:

- In the present version of Fig. 2 a & b, the domain walls (yellow and purple triangles) seem to appear from nowhere as I cannot see the correspondence to the s-SNOM image shown in Fig. 1c).

We understand to point of confusion raised by the Reviewer. To make the correspondence between the domain measured in s-SNOM and photocurrent clearer, we have now sketched the yellow/purple triangles in Fig. 1c to show exactly which domains they correspond to.

- In the main text, the sentence “The black traces in Figure 2a outline the domain wall network of the moiré superlattice measured by optical scattering.” I assume the authors talk about the outlines of the yellow and purple triangles, which only indicate 2 domains. From the wording, I expected many (or all) domains being traced. Also, the wording “black trace” can be misleading as in the figure caption a line trace is used to extract the PC as function of gate voltage. Maybe a clearer description (or more information in the supporting information) would be helpful to better guide the reader through the manuscript.

Overall, the description in the paragraph (line 90-104) was hard to follow as no direct comparison of s-SNOM (domains) and PC images is given.

We have noticed a small editing error that indeed seems to cause confusion. We have reworked this paragraph and the one after (lines 93 – 125) to make these points clearer, including the significance of the black dashed line regarding the gate trace.

3) In line 81, the authors state that “the periodicity varies from 100 nm to 1000 nm in different regions” and attribute this to “local variations of the twist angle”. Is this a reasonable assumption? How much would the twist angle need to vary for this? Would one expect a continuous change in periodicity or some abrupt changes? This could be better elaborated.

It is generally well accepted in the twisted bilayer community that twist-angle variation (usually referred to as “twist-angle disorder”) is apparent in the current state of the art devices and plays a major role in the interpretation of recent experiments. In fact, recent scanning probe experiments made on twisted bilayer graphene (of $\sim 1^\circ$) have mapped out angle inhomogeneities of around 0.2° (Nature 581, 47 (2020)).

For smaller twist-angle devices, as in the case of our minimally twisted bilayer graphene samples, such twist-angle variation has a dramatic effect on the moiré periodicity, which is proportional to $1/\sin(\theta)$. Our observed periodicities 100-1000 nm correspond to $\theta \sim 0.1 - 0.01^\circ$ (see figure below), in agreement with what has been measured by other groups (arXiv:2008.04835), and which is similar to the variations measured in Nature 581, 47 (2020).

Since twist-angle disorder is a rather complex subject in our community, with whole separate studies dedicated to it, we prefer to avoid a detailed discussion of its behaviour in our devices. However, in the revised manuscript we have edited the text in line 82, which now includes the corresponding twist angles and added references supporting our claims of twist-angle inhomogeneity in these structures.

4) The reference section needs to be checked as many mistakes are present:

Ref. 16: page numbers wrong: should be article number 205404

Ref. 17: page numbers wrong: should be article number 241412(R)

Ref. 18: page numbers wrong: should be article number 161406(R)

Ref. 19: page numbers wrong: should be article number 11760

Ref. 24: article number missing: 10699

Ref. 44: journal name missing, Page number wrong: Should be PHYSICAL REVIEW B 90, 075415 (2014)

Ref. 45: journal name missing, ArXiv or published?

Ref. 48: page range useless, article number sufficient

Ref 52: page numbers wrong: should be article number 073110

We thank the Reviewer for careful checking of the references and apologise for the mistakes. We have made corrections accordingly and checked thoroughly that all the relevant articles are correctly cited.

REVIEWERS' COMMENTS

Reviewer #1 (Remarks to the Author):

My concerns have been addressed in the revisions, and I recommend this manuscript to be accepted.

Reviewer #2 (Remarks to the Author):

The manuscript has been significantly improved. Following all of the reviewers suggestions, is contains much better graphical layout and explanantions. It is now suited for publication in Nature Communications, after addressing only two minor remarks.

line 194: "...we find that the domain wall that run..."

I think that the second "that" should be removed, otherwise there seems to be a verb missing in this sentence.

line 253: "...our photocurrent measurements are more sensitive to ray-like modes rather than the first eigenmode typically observed in s-SNOM experiments (ref 41)"

Since this revised version now explains the role of hyperbolic polaritons and especially their "ray-like" propagation in thin slabs of hBN in much more detail, I strongly suggest to also cite the two Nat. Comm. papers (where these ray-like modes have first been directly measured in s-SNOM) in the main text:

S. Dai, et al. Subdiffractive Focusing and Guiding of Polaritonic Rays in a Natural Hyperbolic Material. Nature Communications 2015, 6, 6963.

P. Li, et al. Hyperbolic Phonon-Polaritons in Boron Nitride for Near-Field Optical Imaging and Focusing. Nature Communications 2015, 6, 7507.

Reviewer #1 (Remarks to the Author):

My concerns have been addressed in the revisions, and I recommend this manuscript to be accepted.

Reviewer #2 (Remarks to the Author):

The manuscript has been significantly improved. Following all of the reviewers suggestions, is contains much better graphical layout and explanantions. It is now suited for publication in Nature Communications, after addressing only two minor remarks.

line 194: "...we find that the domain wall that run..."

I think that the second "that" should be removed, otherwise there seems to be a verb missing in this sentence.

line 253: "...our photocurrent measurements are more sensitive to ray-like modes rather than the first eigenmode typically observed in s-SNOM experiments (ref 41)"

Since this revised version now explains the role of hyperbolic polaritons and especially their "ray-like" propagation in thin slabs of hBN in much more detail, I strongly suggest to also cite the two Nat. Comm. papers (where these ray-like modes have first been directly measured in s-SNOM) in the main text:

S. Dai, et al. Subdiffractional Focusing and Guiding of Polaritonic Rays in a Natural Hyperbolic Material. Nature Communications 2015, 6, 6963.

P. Li, et al. Hyperbolic Phonon-Polaritons in Boron Nitride for Near-Field Optical Imaging and Focusing. Nature Communications 2015, 6, 7507.

We are glad to hear that both Reviewers recommend our work to be published in Nature Communications. We thank Reviewer #2 for his suggestions and have implemented them in the manuscript.